# Cloud Platforms for Context-Adaptive Positioning and Localisation in GNSS-Denied Scenarios—A Systematic Review

**DOI:** 10.3390/s22010110

**Published:** 2021-12-24

**Authors:** Darwin Quezada-Gaibor, Joaquín Torres-Sospedra, Jari Nurmi, Yevgeni Koucheryavy, Joaquín Huerta

**Affiliations:** 1Institute of New Imaging Technologies, Universitat Jaume I, 12071 Castellon, Spain; huerta@uji.es; 2Electrical Engineering Unit, Tampere University, 33720 Tampere, Finland; jari.nurmi@tuni.fi (J.N.); evgeny.kucheryavy@tuni.fi (Y.K.); 3ALGORITMI Research Center, Universidade do Minho, 4800-058 Guimaraes, Portugal

**Keywords:** cloud platform, GNSS-denied scenarios, localisation, positioning, systematic review

## Abstract

Cloud Computing and Cloud Platforms have become an essential resource for businesses, due to their advanced capabilities, performance, and functionalities. Data redundancy, scalability, and security, are among the key features offered by cloud platforms. Location-Based Services (LBS) often exploit cloud platforms to host positioning and localisation systems. This paper introduces a systematic review of current positioning platforms for GNSS-denied scenarios. We have undertaken a comprehensive analysis of each component of the positioning and localisation systems, including techniques, protocols, standards, and cloud services used in the state-of-the-art deployments. Furthermore, this paper identifies the limitations of existing solutions, outlining shortcomings in areas that are rarely subjected to scrutiny in existing reviews of indoor positioning, such as computing paradigms, privacy, and fault tolerance. We then examine contributions in the areas of efficient computation, interoperability, positioning, and localisation. Finally, we provide a brief discussion concerning the challenges for cloud platforms based on GNSS-denied scenarios.

## 1. Introduction

Over the last two decades, positioning based on Global Navigation Satellite System (GNSS) constellations has become an indispensable tool for human beings. GNSS-based positioning has been used in many different areas including precision agriculture [1], autonomous navigation [2], and emergency and rescue systems [3]. Despite the advantages of GNSS, its performance tends to degrade in adverse outdoor environments such as urban canyons or areas of particularly dense foliage [4]. In such environments, the positioning error may significantly increase. Furthermore, the signals remain largely unavailable in indoor environments.

Over the last few years, multiple technologies have emerged to provide positioning services to GNSS-denied environments [5,6], for instance, Bluetooth [7], IEEE 802.11 Wireless LAN (Wi-Fi) [8], Pseudolites [9], Ultra Wideband (UWB) [10], ultrasound [11], Visible Light Communication (VLC) [12], and ZigBee [13], among others. The technologies used most frequently for indoor localisation and positioning are Bluetooth Low Energy (BLE) [14,15,16] and Wi-Fi [8,17,18,19] for their availability, cost, usability, and scalability, among others. Although each of these technologies have strengths and weaknesses, none has emerged as an alternative to GNSS for indoor spaces.

Many positioning solutions have been proposed to achieve better performance (e.g., accuracy, availability, reliability, etc.) in indoor environments. Some of these solutions consider Cloud Computing (CC) or similar computing paradigms in order to reduce the power consumption on wearable devices and avoid processing overhead. Furthermore, the CC supports different kinds of services such as Infrastructure as a Service (IaaS), Platform as a Service (PaaS), and Software as a Service (SaaS) [20]. Thus, they are capable of providing services with high performance and quality, low latency, geographic redundancy, and security.

In addition, advances in telecommunications have brought us new and efficient technologies like 5G, which will assist in surpassing the current limitations (e.g., latency, connectivity, data rate, etc.) of wireless communications. This technology includes three main characteristics, first, enhanced Mobile Broadband (eMBB), second, provides Ultra-Reliable Low Latency Communications (URLLC), and third, permits massive Machine Type Communications (mMTC). These components will improve the reliability in communication which is indispensable for mobile devices including wearable and Internet of Things (IoT) devices. This digital transformation arrives alongside changes in the Cloud architecture, using new computing paradigms—CC, Edge Computing (EC), Fog Computing (FC), and Mist Computing (MC)—and applying technologies such as slicing and Network Functions Virtualization (NFV). Helping to ensure high reliable Cloud platforms, as well as to provide reliable Cloud-based indoor positioning and localisation.

Several survey/review articles about indoor positioning and localization have been published in the last few years. However, they mostly emphasised on the technologies, techniques and algorithms for providing positioning indoors [5,6,9,21] and on the role of Indoor Positioning Systems (IPSs) in IoT applications, such as smart cities and smart transportation. Given the rapid development of computing paradigms, devices (smartphones, wearables, IoT, among others) and applications with LBS components, make necessary to perform a systematic review with a new point-of-view oriented to the positioning platforms and the way to interact with them. To the best of our knowledge, this is the first review dealing with indoor positioning and the aforementioned components.

This article thus provides a systematic review of Cloud-based indoor positioning platforms, current challenges and future trends. Moreover, this work introduces the reader to key concepts related to computing paradigms, indoor positioning using mobile devices (e.g., smartwatches, IoT devices, etc.), network protocols and standards used in IPS.

The remainder of this paper is organised as follows. Section 2 gives a general overview of the current surveys and reviews done in the last four years. Section 3 introduces the research methodology used to find the relevant articles together with a description of the systematic review undertaken. Section 4 presents the results from the Systematic Review. Section 5 provides a brief discussion of the main findings, current challenges, and future trends. Section 6 gives a possible validation threats. Finally, Section 7 presents the conclusions of this review.

## 2. Related Work

This section presents a general view of current indoor positioning surveys and their primary objectives. Brena et al. [22] focused their review on the classification of positioning technologies. This review offered an analysis of technologies’ accuracy and coverage area, and discussed the pros and cons of each. The authors successfully identified some future trends, such as the use of crowd-sourcing data to update and enhance localisation systems and the lack of privacy and security in IPS. Similarly, Mendoza-Silva et al. [9] provided an extensive analysis of standard technologies and techniques used in IPS. Moreover, the authors provided a systematic study of IPS solutions published between 2015 and 2019 as well as in [23]. In contrast, Soewito et al. [23] only focused on classifying the articles according to two parameters, accuracy and implementation.

The survey introduced by Kunhoth et al. [24] slightly differs from the previous reviews by including indoor navigation systems and technologies based on computer vision, visible light, and other novel positioning technologies. Hameed and Ahmed [25] based their survey in positioning applications, analysing services such as routing, map data, route selection, accuracy, coverage area, and cost. Ngamakeur et al. [26] offered an analysis of device-free indoor localization and tracking in multi-environments. In this survey, the authors provided a discussion of performance evaluation under the following parameters: accuracy, precision, robustness, scalability, among others. This review also focused their analysis on techniques for localization and tracking, for instance, human detection, identification, and counting. Simões et al. [21] focused their work on positioning technologies for visually impaired people, this work also provides a classification of current technologies and techniques in similar way that [5,6,9,26].

The increased interest in IoT and ubiquitous connectivity promote the development of techniques and methods for IoT devices. These methods and techniques have been examined in several surveys. For instance, Khelifi et al. [5] emphasized the importance of localisation systems in cutting-edge IoT applications such as smart cities. In this survey, the authors provided a new classification of Indoor Location System (ILS) as distributed, centralized, and interactive systems. Additionally, the authors stressed out the need of integrating green computing in services development, i.e., developing computationally efficient IPSs without sacrificing accuracy. Shit et al. [27] conducted a review based on Location of Things (LoT), providing an extensive analysis of current localisation techniques in IoT infrastructure. In contrast with a previous work, this review included in their analysis techniques based on machine learning.

Zafari et al. [6] analysed IPS/ILS in terms of energy efficiency, cost, range, time response (latency), availability and other parameters. Moreover, they included a review of current systems and foresee the future challenges. Liu et al. [28] focused their survey in Channel State Information (CSI)-based IPS methods and techniques. This work provided an extended analysis of algorithms, including deterministic, probabilistic and Machine Learning (ML) algorithms. Obeidat et al. [29] also placed special emphasis in current techniques, technologies, and algorithms used in IPSs, as well as their pros and cons. In other reviews [21,30], the authors discussed particular topics related to indoor positioning and localisation. For example, Saeed et al. [30] discussed the Multidimensional Scaling (MDS) based on indoor and outdoor positioning techniques. The authors analysed the application of MDS on different networks, such as mobile networks (e.g., 5G), applications, and even different environments.

In addition to the cited reviews and surveys, there are many other research works highly relevant for indoor positioning and localisation. Some of them provide an extensive and deep analysis of very specific topics as, for instance, the evaluation of pedestrian localisation systems [31]. Bousdar Ahmed et al. [31] provided discussion on benchmarking datasets, simulators, record & replay, and the theoretical performance limits. Despite those topics being relevant for indoor positioning evaluation, they are out of the scope of this review, which is focused on platforms providing positioning services.

Table 1 introduces a summary of the comparison between the current surveys and reviews from 2017 to 2021. Most of the surveys offered a comprehensive discussion of technologies, techniques, and methods used in IPS. Additionally, some authors provided a brief discussion of current challenges from the point of view of indoor positioning technologies, techniques, environment, devices, and security and privacy. For instance, there are challenges related to multipath effects, line of sight, signal interference in IoT networks, energy efficiency, synchronization, crowded environments, among others. However, many authors agree that the environment is one of the principal challenges for IPS given its high complexity [5,6,22]. In either case, the positioning system have to be carefully evaluated under different conditions in order to test its performance, robustness, flexibility, precision, and accuracy [26].

As can be observed in Table 1, mobile devices have become robust enough to process indoor positioning information. Nevertheless, there are still multiple challenges related to energy consumption, cloud computing and IPS, battery life, security, and privacy that have not been fully dealt with on them.

Computing paradigms (CC, EC, FC, MC) bring multiple advantages, such as high computing and storage resources, and extend services provided by current applications, and allow straightforward integration between systems. These computing paradigms have not been the case of study in recent surveys and reviews of indoor positioning and localization. Similarly, standards and protocols have not been part of these reviews, in spite of being crucial for current IPS/ILS.

## 3. Research Method

This section introduces the procedure and methodology used to identify studies relevant to this systematic review. The methodology has been selected for its clear procedure which can be easily reproduced by other researchers, to comprehensively analyse the published research, identify current trends, and detect the unexplored research lines on a particular topic.

As part of the systematic review we used the Preferred Reporting Items for Systematic reviews and Meta-Analyses (PRISMA) guidelines [32], consisting of a twenty-seven item checklist together with a flow diagram divided into four parts (identification, screening, eligibility, and included).

### 3.1. Research Questions

Setting the right research questions is a key stage of any systematic review, as it is paramount importance to identify the main objectives of the analysis. We conducted this review with the following main research question (MRQ):**MRQ** What are the possible gaps or issues in Cloud Platforms for positioning and navigation in GNSS-denied environments?

This main question is generic, therefore, we broke it down into the following specific research questions (RQ):**RQ1** Are the main computing paradigms used in current indoor positioning platforms?This research question permits to identify if the current indoor positioning platforms are using the main computing paradigms. This question allows us to determine future trends regarding indoor positioning platforms and computing paradigms.**RQ2** What network protocols do the current platforms use to provide reliable services?This question addresses the need to know which network protocols are used in the current indoor positioning platforms. Additionally, this research question helps to identify strengths and weaknesses of the used communication protocols in the scope of indoor positioning.**RQ3** Do the current platforms permit heterogeneous positioning technologies for GNSS-denied scenarios?This research question allows us to determine the current platforms’ flexibility to support diverse position technologies.**RQ4** Do the current platforms adapt to different scenarios?As the diversity of indoor scenarios are currently considered a main challenge for providing positioning indoors, this research question helps to identify the limitations of the current systems.**RQ5** What were the improvements done in similar studies?This research question aims to identify the main contribution of the studies analysed and the current challenges in indoor positioning platforms.**RQ6** How is the standardization aspect dealt with on different platforms?Standardization is key to providing a more reliable and high-quality indoor positioning platform. Thus, this research question aims to identify if the current systems consider the existing standards for IPS in different dimensions.

### 3.2. Keywords

The number of research studies relating to indoor positioning has increased exponentially over the years. Thus, it is important to define clear search queries and strategies to pinpoint the most relevant publications related to the topic of this systematic review. We therefore proceed with the identification of keywords related to the research topic and its objectives. The keywords were chosen according to the infrastructure, environment, and system.

Table 2 shows the keywords we have chosen in the research process. The wildcard pattern (* in the queries) means any number of characters. In our queries we introduced them to identify some related concepts that have the same prefix (e.g., position, positioning, positions, etc.).

### 3.3. Query

Once keywords are defined, the next step is to give form to the search query which will be used to find relevant works. This query was used in two well-known search engines *Web Of Science* and *SCOPUS* using the advanced search of each site.

*Web Of Science* Query:


TS=((((( cloud OR edge OR fog OR mist ) AND ( computing OR paradigm )



OR platform ) AND ( indoor* OR gnss-denied ) AND ( position* OR location



OR localisation ))) Timespan
: 2015-2021


*SCOPUS* Query:


TITLE-ABS-KEY((((( cloud OR edge OR fog OR mist ) AND ( computing



OR paradigm )) OR platform ) AND ( indoor* OR gnss-denied )



AND ( position* OR location OR localisation )))



AND ( LIMIT-TO ( PUBYEAR , 2021 ) OR LIMIT-TO ( PUBYEAR , 2020 )



OR LIMIT-TO ( PUBYEAR , 2019 ) OR LIMIT-TO ( PUBYEAR , 2018 )



OR LIMIT-TO ( PUBYEAR , 2017 ) OR LIMIT-TO ( PUBYEAR , 2016 )



OR LIMIT-TO ( PUBYEAR , 2015 ))


Both queries will return a list of works (conference papers, journal papers, books, among others). Despite the fact that many of the retrieved records are related to the research topic, some of them do not meet the requirements of this review.

### 3.4. Study Selection

The results retrieved from the previous search has to go through a process described in PRISMA diagram and checklist document. This step includes identifying all relevant studies, removing duplicate records, and defining inclusion and exclusion criteria. Those criteria form the basis for the ultimate decision as to which works are included in the qualitative and quantitative synthesis.

#### 3.4.1. Stage 1: Identification

Both SCOPUS and Web of Science are important databases which index works from different sources such as IEEExplore, SpringerLink, ArXiv, Wiley Online Library, and others. Merging the results from both data sets drives to duplicate records which must be removed. Thus, the retrieved records, along with their bibliography and metadata, are stored using reference manager software. This software is used to remove duplicate records and classify the studies obtained from the search engines.

#### 3.4.2. Stage 2: Screening and Selection Criteria

Once we have removed duplicate records, we obtain several unique registries, which have to be filtered to obtain only relevant publications for this review. Thus, we defined the following inclusion criteria (IC) and exclusion criteria (EC).

**IC1** Full research works written in English**IC2** Research works dealing with platforms supporting positioning**EC1** Works not dealing with any computing paradigm (e.g., Cloud computing) or GNSS-denied scenarios**EC2** Works not published in peer-reviewed international journals or conference proceedings**EC3** Studies not dealing with wearable devices (we consider smartphones as wearable devices)

Although the term ‘Cloud computing’ has been used since 2007 [33], and indoor positioning has been studied for many years, we limited this review to articles published in the last 7 years (from 2015). The decision to select material published between 2015 and 2021 is based on the increase in demand for Cloud services since 2015. According to CISCO annual reports, we can see an evident increase of Internet usage in the last 6 years. Moreover, data centres’ traffic has increased three-fold from 2.2 Zettabytes in 2015 to 7.1 Zettabytes in 2020 [34,35,36,37], meaning that users and companies are increasingly utilising services hosted in the Cloud.

In order to select only those works that fulfil all the requirements established in the *IC* and *EC*, we proceeded with the manual revision of titles and abstracts for their subsequent tagging with *ACCEPTED* for accepted articles and *REJECTED* + *EC* for rejected records. Overall, we selected around 8% of the total number of studies obtained in the previous stage.

#### 3.4.3. Stage 3: Eligibility

In this stage, we carefully read each remaining study under the consideration of the main objective of this review and the established *IC* and *EC*. If the article reviewed does not accomplish the requirements established in previous steps, it is excluded from this work.

#### 3.4.4. Stage 4: Included

The studies are categorized according to their conclusions and contributions in the research field (Cloud-based Indoor Positioning and Localisation). This step is the last filter in order to select only relevant publications for this review.

### 3.5. Main Figures for the PRISMA Process in the Current Review

Figure 1 shows the flow diagram and the results after following the whole process. A total of 2369 works (articles, papers, magazines, book) were retrieved from *SCOPUS* (1642) and *Web Of Science* (1458) search engines after executing the defined queries.

First, 2085 unique records were obtained after removing duplicate records. Despite the overlap between the two search engines, there has been a significant number of work published in a conference or a journal which was only indexed by one database. However, this list of research works included mostly unrelated studies—either not dealing with computing paradigms or GNSS-denied environments—which were filtered during the initial screening, that means almost 96% of retrieved works. Furthermore, the works not dealing with wearable devices, representing almost 11% of retrieved works, were also filtered out.

A total of 158 works were selected after screening the inclusion/exclusion criteria on the titles and abstracts. Finally, after a full article assessment, only 83 studies fulfilled all the requirements for this systematic review and, therefore, we included them in our comprehensive analysis.

### 3.6. Overview of the Selected Studies

Although the search queries provided 2085 works, only 83 of them fulfilled all the criteria established in this work and were, therefore, analysed (see Figure 1). The temporal distribution of the selected works is shown in Figure 2, where the type of article is also differentiated.

Most of the selected articles were published in conferences with 49 research works, followed by 33 studies in different journals (e.g., IEEE, ACM and SpringerLink) and just one magazine article. In recent years (2019 and 2021), the works published in journals have been significantly increasing, reaching a ratio conference/journal of 1:0.8 in 2020. In 2021, the number of articles has decreased compared to the previous years (last update: November 2021). Overall, there is an upward trend in the last seven years.

The numbers of 2021 may not be fully conclusive as the databases were still indexing papers from 2021 after we performed the search. Moreover, the exceptional events which occurred during 2020–2021 may have negatively impacted on the research in the studied topic. Strict lockdowns around the world have reduced empirical testing and research, many conferences on related topics have been cancelled, and some research teams have focused on side topics such as contact tracing.

### 3.7. Data Extraction

During this process, we collected all the relevant information from each of the 83 selected studies. This information includes computing paradigms used in current indoor positioning platforms (RQ1), network protocols (RQ2), position technologies (RQ3), testing and deployment areas (RQ4), main goals and improvements achieved by the authors (RQ5), and the standards used by researchers in their systems (RQ6). The main outputs of this process are reflected in Section 4, whereas the main features extracted from the analysis can be found in Table A1 included in the Appendix A.

## 4. Results

This section will analyse the key information extracted from the 83 selected studies (see Section 3.7) in order to answer the five defined research questions (see Section 3.1).

### 4.1. Computing Paradigms Used in Current Indoor Positioning Platforms (RQ1)

The exponential growth of devices connected to the Internet has led to the development of new computing paradigms providing ubiquitous computing, security & privacy, and low latency, among others. These computing paradigms are currently used in IPSs to improve the Location-Based Services (LBS) in terms of Quality of Service (QoS) and Quality of Experience (QoE) to the end-user. Despite IPSs having been studied for decades, the new computer paradigms have been only exploited over the last few years (see Figure 2).

We have identified six computing paradigms in the 68 analysed research studies, which are described in the following paragraphs.

#### 4.1.1. Cloud Computing (CC)

This computing paradigm provides different services by extending the computing resources, storage and network capabilities to the Internet [38]. These services are offered through remote resources located in large data centres. They are characterized by their high computational and storage resources [39], on-demand services, redundancy, among relevant features. Here we can distinguish three main models listed by the National Institute of Standards and Technologies (NIST), which are SaaS, PaaS, and IaaS. Thus, the management tasks can be distributed between the IPS developers and the Cloud provider.

In SaaS, everything is managed by the service provider. For instance, the well-known Google platform provides many different services (e.g., storage, monitoring, etc.) without any technical effort from the user’s side.

In PaaS, the application and data are managed by the user [38] but the remaining services are managed by the Cloud provider. For example, IBM provides these services for deploying native applications on the Cloud.

IaaS has four services—virtualization, storage, server, and networking—maintained by the Cloud provider. For instance, Microsoft Azure offers servers, storage, networking, and data centres managed by itself. These primary models are used as a basis for a positioning/localisation service [40,41], which the architecture is based on PaaS.

There are other models or Cloud services proposed by current research studies, such as Indoor Navigation as a Service (iNaaS) [42]. As the name implies, iNaaS provides navigation services to other indoor applications. Thus, the beacon’s information collected by the mobile device (i.e., the list of Received Signal Strength Indicator (RSSI) and identifiers) is sent to the Cloud service in order to get the navigation information. Moreover, positioning evaluation services are also deployed on the Cloud [43], so developers can test their data sets and their indoor positioning solutions under certain standards such as the ISO/EC 18305 standard “Test and Evaluation of localisation and Tracking Systems” [44].

Despite the wide range of Cloud services and functionalities, most of the research studies used Cloud Computing for its high computational and storage capabilities. Many positioning algorithms are time- and resource-consuming. Developers therefore often prefer to migrate these processes to the Cloud to run positioning/localisation, navigation, tracking algorithms [7,45,46,47,48,49]. Moreover, when the IPSs have to manage multiple and large environments, low profile devices are unable to store that level of information. The Cloud is therefore used to store indoor positioning information such as Wi-Fi radio maps, Point-of-Interests (POIs), map layers [50,51,52,53].

Additionally, developers, companies, or researchers can opt for using Private, Public, Community, or Hybrid Cloud. The services deployed in the public Cloud can be accessible for everyone that require them, for example, Google services (e.g., Gmail, Google drive, etc.) Private Cloud infrastructure is exclusively for a specific entity (single organization) and its users, similar to on-premises infrastructure. Community Cloud is characterized by their sharing of services between different organizations. And, hybrid Cloud is simply the combination of the aforementioned Cloud types [38].

To deploy IPS on the Cloud allows the provision of more stable and reliable services. Hence, the Cloud providers work under a Service Level Agreement (SLA), and their data centres must accomplish high-quality standards (TIER, ANSI/BICSI 002-2019, etc.) and fulfil different requirements such as geo-redundancy, high availability, and power outage protection in order to obtain the corresponding certifications.

Figure 3a shows the basic schema of CC, where end-devices are connected to the Cloud through the Internet.

#### 4.1.2. Mobile Cloud Computing (MCC)

This paradigm combines CC and MC. The main purpose is to move the most demanded computational process from the mobile/wearable devices to the Cloud in order to optimize the use of local resources [54]. It avoids processing large amounts of data in the user’s device. However, this kind of implementation is also susceptible to different issues such as delays in communication, security and privacy, etc. Moreover, it cannot work offline, being Internet connectivity a must. Despite its limitations, Mobile Cloud Computing (MCC) is widely used because many services are provided through mobile devices. Therefore, many researchers have implemented their platforms by using this computing paradigm in order to provide commercial solutions, open-source systems [55], and academic research platforms [56].

Due to the mobility required for indoor positioning applications, the use of mobile devices or wearables is required. However, many mobile devices are resource-constrained, therefore, some positioning techniques cannot be executed on those devices. According to Khan et al. [57], MCC has four main objectives related to *performance*, *energy-consumption*, *constrain devices*, and *multi-objective MCC model*.

For instance, Huang et al. [58] proposed a new IPS, which offer a low energy consumption without affecting the position accuracy. The low energy consumption is obtained by dividing the environment into multiple sub-areas and applying state controls in each region. Noreikis et al. [59] developed an efficient Augmented Reality (AR) navigation system, which aims to provide an efficient service in terms of performance and energy consumption. Given that vision-based indoor navigation systems consume more energy than other technologies, such as BLE, Wi-Fi, or UWB, it is necessary to run some tasks outside of the mobile device. Therefore, in [59], the computing-intensive process was migrated to the Cloud to offer an efficient navigation system. Moreover, the authors proposed a new algorithm to reduce the image sample frequency, which maintains the position accuracy. The researchers improve the system performance and reduce the energy consumption, considering these systems as *multi-objective MCC model*.

Similarly, Silva and Wimalaratne [11] designed a wearable belt with vision and sonar sensors to provide navigation services for people who require assistance devices, such as visually impaired persons. In this case, the authors use the advantages of Google Cloud to process the environment’s images, allowing the process to be offloaded to the Cloud.

Figure 3b shows the general architecture of MCC used in the analysed studies, where the mobile devices (including smartphones, wearables, and IoT devices) are connected to the Internet through wireless communications such as Wi-Fi or mobile network services. In order to determine the user position, the mobile device requires the assistance of positioning technologies (explained in detail in Section 4.3), such as Wi-Fi, UWB, and BLE. Sometimes the researcher combines these positioning technologies with Inertial Measurement Units (IMU) sensors (accelerometer, gyroscope, etc.) in a tight or loose coupling sensor fusion approach.

#### 4.1.3. Fog Computing (FC)

In the last decade, the number of wearable and IoT devices on the market has increased exponentially. Taking into account that many of those devices use position, location, navigation, or tracking services, the use of FC confers an advantage for IPS. Hence, FC was designed to decentralize the computational load, provide low latency for real-time applications [16] and massive device connectivity [60]. This paradigm is increasingly utilized for latency aware services, such as indoor navigation services.

In order to exploit the advantages of this computing paradigm, Sciarrone et al. [61] implemented the Smart Probabilistic FingerPrinting (P-FP) algorithm in the FC platform. It allowed to reduce the computational load, and therefore, the energy consumption in the mobile device. When the user’s device falls below a certain battery level, the computation of the P-FP algorithm is distributed into the near devices, significantly reducing the energy consumption (the energy saving was more than 80%).

As mentioned earlier, FC permits massive device connectivity. In this fashion, Li et al. [62] make the BeDI repository system a reliable and scalable Fog implementation. This system was developed to give support to positioning systems, providing fast response to hundreds of thousands of devices. Similarly, in crowdsourcing data collection, many devices are connected to the IPS/ILS sending a significant amount of data to the platform waiting for a fast response. However, the service performance may be degraded due to the high data traffic in the network. To avoid it, Li et al. [63] used FC to reduce the computational load and store historical information. Additionally, due to the low latency provided by this computing paradigm, it is suitable for real-time indoor positioning applications such as the application proposed by Pesic et al. [16].

#### 4.1.4. Mist Computing

Although Mist computing is a relatively new computing paradigm, it has already been used in many architectures in cooperation with other computing paradigms. This computing layer is located close to the endpoints. Its main approach is to extend FC capabilities with the help of IoT devices, which is why it is also known as “Things Computing” [38]. However, its computational and storage resources are greater than FC resources. In the analysed studies this computing paradigm is combined with FC [64] (see Figure 3g). Battistoni et al. [64] set a mesh network of Mist nodes, where different process take place, such as data analysis, pattern search, predictions, etc. Some of these tasks use ML techniques applied to indoor positioning to determine the user position and occupancy in challenge environments, among others.

#### 4.1.5. Edge Computing

This computing layer is geographically close to the end-user, and it is characterized for its low latency, real-time data access, storage, and computational capabilities (small data centres) [38,60,65,66]. It helps to provide scalable networks and platforms. In the current studies, we find two types of architectures that use EC. The first one is device-EC (see Figure 3d), where certain processes are migrated from the mobile device to the Edge. For instance, ref. [67] used the EC to estimate the user location, update the probability positions from the Received Signal Strength (RSS) measurements obtained from the user device. Additionally, the EC is in charge of storing the data collected in the local database. The authors mention some advantages of using this computing paradigm, first, lower latency than using CC, an improved position estimation, and its support for other analysis based on the user location.

Ben Ali et al. [68] distributed the computational load of their system between the mobile device and the EC. Being Edge-SLAM a system based on visual technology, the computational load that has to support the mobile device is greater than other technologies, as we mentioned before. Thus, only the tracking process is running in the user device, and the remaining processes are executed in the EC. As a result, the authors reduced the mobile device’s energy consumption with a minimal decrease in the map accuracy and the trajectory taken by the user. Fazio et al. [69] used the EC to process the navigation information, moreover, they analyse the impact of ML on the EC.

Given the massive device connectivity to the EC, in order to process positioning information, it can be susceptible to multiple attacks. This can cause severe security & privacy issues, which have to to be minimized as much as possible. In view of these issues, Zhang et al. [70] proposed a novel method based on the differential privacy-preserving mechanism. This proposed lightweight mechanism not only guarantees the privacy during the training stage of Wi-Fi fingerprinting, but also the accuracy of the indoor localisation. Similarly, Liu and Yan [71] implement a security layer to the EC, but it is addressed to outsourced ILS. The authors mentioned that similar implementation for the CC are not suitable for this paradigm, and therefore, they used the concept of backdoor in ML to design a new verification schema.

The second architecture (see Figure 3h) combines the advantages of CC and EC. In such a way, the heavy processes are executed either on the Edge or the Cloud. For instance, Liu et al. [72] process the tracking information on the Edge and the Cloud is used to find the best path for the user (navigation). This system is specially designed for contact tracing applications, such as the applications used for social distance. In the same fashion, Liu et al. [72] process location information on the Edge, and store the processed data in the Cloud.

#### 4.1.6. Multi-Access Edge Computing

This paradigm uses Multi-access Edge Computing (MEC) servers near the mobile base station, which provide multiple services to the end-user (including enterprise customers) such as IoT, AR, catching, location services, among others [38,60]. Current IPS/ILS use two types of architectures based on MEC, the first one is mobile-MEC (see Figure 3e) and the second one combines MEC and CC (see Figure 3i). This paradigm is also involved in the deployment of the new mobile technology 5G by offloading certain process on it [73].

As an example of the first architecture (mobile-MEC), Horsmanheimo et al. [74] used MEC to run positioning services. Moreover, this proposal system has been built by using the 5G testbed infrastructure provided by Espoo. Correspondingly, Li et al. [75] pointed out the need of using servers with high capabilities to run ML algorithms. Additionally, the authors mentioned that the use of local server would change progressively with the emergence of MEC.

To illustrate the second architecture, Carrera V. et al. [76] used both the CC and MEC. MEC is used in order to process all the information related to the tracking of users and provide real-time localisation. Meanwhile, the Cloud is used to store the location information for future analysis. As a result, the authors obtained a tracking error of 0.44 m.

### 4.2. Network Protocols Used in Current Cloud-Based Indoor Positioning Platforms (RQ2)

Network protocols are one of the principal components to guarantee the interoperability and communication between systems and devices regardless of the vendor and structure. These protocols operate in different layers of the Open System Interconnection (OSI) model or similar models, exchanging information between upper and lower layers or vice versa.

These protocols can be either proprietary or open protocols. Both have pros and cons that must be carefully analysed according to the needs of the application. In this section, we will report the protocols used in the analysed studies. We divided the used protocols into four main groups, communication, security, IoT protocols and other protocols used in the current studies.

#### 4.2.1. Communication Protocols

Cloud-based indoor positioning systems can use different protocols to establish and maintain the communication between the Cloud (or other computing paradigms) and the user device. Given the heterogeneity of devices used by both users and service providers, the use of protocols is essential to avoid problems in the communication and provide a common language between them. Nowadays, many protocols have been developed for new, reliable, and efficient communications. However, the traditional protocols are still widely used in multiple services.

As part of the TCP/IP protocol suite, we have *HyperText Transfer Protocol (HTTP)/ Hypertext Transfer Protocol Secure (HTTPS)*, which is an application layer protocol (OSI model). HTTP is responsible for exchanging information between the web client and the web server. In the studies analysed, developers or researchers use the REpresentational State Transfer (REST) architecture style to exchange data between the client and the server over HTTP [43,45,54,55,77,78,79,80], but REST is not restricted to this protocol. The use of REST architecture style allows establishing a stateless communication by using Uniform Resource Identifiers (URIs) and HTTP verbs (get, post, put, and delete) [81]. For instance, Ref. [82] use a REST Application Programming Interface (API) for a seamless connection between the client and the web server. Thus, when the user reaches a place, the location is updated by using the REST API service.

Another example is given in [80], where the authors propose a new shopping platform based on IoT solutions. According to the diagram presented by the authors, the communication between the IoT devices and the back end is done using the HTTP protocol and REST architecture style. Current applications or services which implement REST architecture style (RESTful web services and RESTful APIs) are commonly used in many applications. Thereby, Raspopoulos et al. [79] mentioned that the use of RESTful API and other components allow a straightforward development and integration between applications. However, some researchers have migrated from HTTP protocol to *websocket* protocol [76,83], given that it overcomes some common problems of using HTTP as a transport protocol (e.g., multiple underlying Transport Control Protocol (TCP) connections between the client and server, constant monitoring of the client connection to track replies, among others). This protocol was used to establish the communication between the client and the positioning server.

As part of the communication protocols, *OpenFlow* is also used in the current studies. It represents the communication between the OpenFlow switch and the controller by exchanging three types of messages, controller-to-switch, symmetric, and synchronous [84]. Guo et al. [85] installed OpenWrt and Open vSwitch on wireless routers, turning them into virtualised routers. They are used to communicate the virtualised routers with the Fog layer through the OpenFlow protocol.

Another protocol used in the reviewed articles is *OBject EXchange (OBEX)*. OBEX protocol thus allows to exchange data and control messages between devices through a wireless communication [86]. For instance, this protocol is used in [62] to exchange the position information between the Lbeacons and mobile devices.

The *User Datagram Protocol (UDP)* protocol is commonly used as well in indoor positioning platforms. However, unlike of TCP, UDP is not connection-oriented. Given that wearables (wristbands, smartwatches, smart glasses, etc.) and IoT devices are energy-constrained, they require lightweight protocols to transmit data [87]. Thus, UDP is rapidly growing in popularity between these devices. For example, Chen et al. [88] use the UDP protocol to transmit the collected data (RSS and Media Access Control (MAC) address) from the wristband to the server where the user position is estimated.

#### 4.2.2. Security Protocols

Nowadays, security and privacy issues are the biggest concerns both for the academy and the industry. These are also an issue for indoor positioning applications, especially when they are on a Cloud-based platform. Given that some sensitive information of the users is sent to the Cloud in order to estimate the user position, it can be intercepted by unauthorized people or suffer from location spoofing attacks [89,90]. Moreover, wireless communications are susceptible to multiple attacks, in the case of indoor positioning, the attacker can emit corrupted RSS values in order to affect the process to estimate the user’s position.

To ensure the secure communication between the user device and the Cloud, many researchers have implemented the security protocols in their communications. For instance, Biehl et al. [91] used the *Transport Layer Security (TLS)* protocol to ensure the communication between the client and the Cloud server. In such a way, the authors are able to transmit the user position through a secure channel by encrypting the communication, making it difficult for an unauthorized third party to understand the transmitted data. TLS is a cartographic protocol which are also known as SSL/TLS protocol, where Secure Sockets Layer (SSL) is the previous version of TLS.

In order to guarantee confidentiality, security, and integrity, TLS protocol use two extra protocols the TLS handshake protocol and TLS record protocol. TLS record protocol is responsible for the confidentiality by using symmetric key cryptography and a keyed Message Authentication Checksum (MAC), whereas TLS handshake protocol carried out the session negotiation [92].

#### 4.2.3. IoT Protocols

Given the high proliferation of IoT devices in the industry that use positioning services, it is necessary to have protocols which provide reliable communication between these devices. This is the case of *Message Queuing Telemetry Transport (MQTT)*, which is a Machine to Machine (M2M) lightweight IoT protocol. It has three components—the broker, subscriber, and publisher [81]. The publisher is any device (client) that sends messages through the MQTT broker, and the subscriber is the client connected to the broker interested in a specific message. Here, the broker is responsible for dispatching the messages between the clients. This protocol is widely used by devices that require low power consumption, narrow bandwidth, and small data package [7,93].

MQTT has been used by [7,16,46,56,64,94,95] in their indoor positioning/localisation platforms. For instance, Chatzimichail et al. [95] used this protocol to perform encrypted real-time communication between the modules. Navya et al. [94] used to communicate the edge nodes with the main node. In [7], the MQTT dispatches the RSS values collected by the clients to upper layers subscribed to the broker.

*Extensible Messaging and Presence Protocol (XMPP)* is another IoT protocol used in the retrieved studies from the systematic review. XMPP is an open and decentralized messaging protocol that allows exchanging messages between the clients regardless of the operating system [81]. Kulshrestha et al. [96] used this protocol to provide reliable communication between the portable sensing units (PSU) and the XMPP server and vice versa in order to compute the top-k queries.

#### 4.2.4. Other Protocols

These protocols are used to support particular services provided by the ILS, for instance, the Location-to-Service Translation Protocol (LoST) protocol, which is used to determine or locate the nearest Public Safety Answering Point (PSAP) according to the geographical position of the user [97,98].

### 4.3. Do the Current Platforms Permit Heterogeneous Positioning Technologies for GNSS-Denied Scenarios? (RQ3)

In order to determine the position in GNSS-denied scenarios, it is necessary to combine technologies, techniques, and algorithms. These combinations will help us to acquire different levels of accuracy in the position estimation. However, the component selection depends on the precision required in a specific environment and its complexity. While the robotic industry requires a highly accurate position, a school does not need high position accuracy to find classrooms. Nowadays, there are multiple technologies used for positioning and localisation based on radio frequency, magnetic field, light, sound, infrared, sensor fusion, among others. All of these technologies provide a different range of precision. Thus, some of these technologies offer centimetre-level accuracy while others still offer a few meters of accuracy [72,99,100].

Technologies featured in the studies analysed have been classified into five main categories; radio frequency (RF) technologies, inertial sensors, computer vision, sound, magnetic field, and optical technologies. Additionally, techniques and algorithms used in each study will be listed in this section. Many of these techniques have been analysed in the surveys and reviews cited in Section 2.

#### 4.3.1. Radio Frequency Technologies

Nowadays, many technologies based on radio frequency have been deployed in multiple environments, both indoor and outdoor. Although some of these technologies were not designed for positioning and localisation services, they are currently used to accomplish this, for instance, Wi-Fi.

##### IEEE 802.11 Wireless LAN (Wi-Fi)

Wi-Fi is part of the IEEE 802.11 standards family. Many indoor positioning/localisation solutions are based on this technology for its low cost, and because it is already deployed in many places, both indoor and outdoor [101]. Moreover, Wi-Fi is supported by a large variety of devices such as mobile phones, laptops, among others [6,45].

In order to estimate the user position by using Wi-Fi technology, researchers and developers use different techniques such as fingerprinting [73,102], which is based on the RSS measurements. Fingerprinting is divided into two phases, the offline phase, where the radio map is formed from the collected RSS values and the online phase, where the incoming fingerprint is compared with the fingerprints in the radio map. From the RSS values we can also determine the user position by using signal propagation models (e.g., Path Loss Model [75]).

The performance of Wi-Fi fingerprinting is reduced when there are thousands of RSS values and multiple floors and buildings in the dataset. That is why some authors have proposed to reduce the radio map [102] or taking only a portion of the radio map [103] to improve the performance on huge Wi-Fi fingerprinting datasets. Another technique used in analysed studies along with Wi-Fi is Time of Arrival (ToA), which uses the signal time of flight to estimate the distance between the transmitter and the receiver. For instance, Lemic et al. [8] stored row data of both ToA and Wi-Fi fingerprints in the dataset, which were used to evaluate indoor positioning algorithms.

As a result of using Wi-Fi technology in indoor positioning applications, the authors acquired different location accuracy levels. For instance, Pericleous et al. [103] got an average location error of 1 m (approximately) on their indoor localisation service. Chen et al. [88] acquired a mean positioning error of 2 m by combining fuzzy logic and genetic algorithms.

##### Bluetooth

Bluetooth is used for short-range communication between devices, becoming one of the most used technologies worldwide, for instance, in wireless mouse, headphone, among others. The last versions of Bluetooth (v4.0 and v5.0) provide low energy consumption, which is why they have been broadly used in power-constraint devices. Unlike the previous versions, Bluetooth protocol 5.0 supports different data rates (125 kbps, 500 kbps, 1 Mbps, 2 Mbps), and it is capable of transmitting information up to 400 m [104].

Li et al. [7] mentioned in their study that Bluetooth 5.0 will help to overcome two main problems associated with the location prediction done by traditional propagation models. Thus, Bluetooth 5.0 will improve the accuracy and the system’s stability.

Currently, BLE is being used in many indoor positioning platforms both with wearables and IoT devices [105]. Generally, the researchers implement IPS/ILS based on BLE by deploying beacons, ibeacons [50], or custom BLE devices in the environment [106]. However, it is not always needed to deploy beacons in the environment. For example, if we need to determine the distance between two individuals, it is only required the mobile user devices equipped with Bluetooth, i.e., using the RSSI of BLE advertisements for measuring relative distances.

There are two protocols based on BLE, the first one was developed by Apple, namely iBeacon, and the second is the Google version, known as Eddystone. These protocols were designed to provide proximity services by using four regions *immediate* ( 0 m ≤ distance ≤ 1 m), *near* ( 1 m < distance ≤ 3 m), *far* ( 3 m < distance ≤ 50 m), and *unknown* (device not ranged) region [6,82]. However, the signal might be affected by the environment, and therefore, the proximity accuracy as well. Thus, the use of filtering techniques (e.g., Kalman filter (KF), Winsorization, Trimmed Mean, etc.) is necessary to reduce the errors inherent to this technology [80,95,107].

The proximity is based on the RSS values, which can be calibrated by the user. Similarly, the user can set the advertising package frequency in order to reduce or increase the time between the broadcast of each message. The iBeacon advertising message contains a constant preamble of 9 B long (e.g., flags, type, companyID, etc.), UUID 16 B long, minor, and major 2 B long each, and finally, the measured power RSSI 1 B long [50,82].

Many indoor positioning techniques used for Wi-Fi can be also used for Bluetooth, such as fingerprinting, ToA, among others. A common method used with Bluetooth is trilateration [95,105,107], it consists of determining the position by using three reference points and the distance. Multilateration relies on the time difference in the arrival of signals to various base nodes, in this case access points. We also have triangulation, which unlike trilateration, uses the angles instead of the distance to determine a point of position [9,22].

The accuracy achieved by using BLE ranges from a few centimetres to a few metres. For instance, [46] obtained less than 2.6 m in an area of 600 m^2^ and 19 beacons deployed on it. Ref. [7] achieved 0.86 m location accuracy in an area of 12 m ×  16 m and combining KF, Long short-term memory (LSTM) + Tri (Multi-Weighted-Centroid).

##### Ultra Wideband (UWB)

UWB is a RF technology with wide spectrum of frequency and high bandwidth ( −10 dB bandwidth > 500 MHz and centre frequency > 2.5 GHz) [10]. There are two specifications of UWB which are small centre frequency and relative large bandwidth [5], these two characteristics make UWB suitable for different purposes such as to go through different materials [6]. Moreover, it may operate with high data rate and a very low power level.

Given the high-accuracy provided by this technology, it has been used in some indoor positioning solution both open-source [55] and commercial solutions (e.g., Insoft). For example, Barua et al. [10] used UWB technology due to its high accuracy in real-time. As this system is designed for people with dementia, high accuracy in real time is a must, that is why they combined UWB + EC + ML (Support Vector Machine (SVM) + *k*-Nearest Neighbor (*k*-NN)) in order to analyse patterns in their behaviour linked to their mobility.

Carrera V. et al. [76] mentioned the robustness of UWB against multi-path effects, a standout feature when compared to other RF technologies. In this study, the authors combined UWB and MEC. Similar to Barua et al. [10], three ML algorithms were used—KStart, Multilayer Perceptron (MLP), and CART. As a result, this system (InTrack) acquired an average tracking of 0.59 m, which is better than commercial solutions according to the authors.

As we can see, the authors prefer UWB for highly accurate and reliable solutions, where real time is required. Along with it, the use of ML algorithms were used in these applications. Thus, two well-known algorithms were used, including *k*-NN, which is used for classification and regression problems. *k*-NN searches the nearest observation in order to predict or classify the new observation based on the other observations. SVM is also used for classification and regression but it uses hyperplanes [10]. CART is the classification and regression tree algorithm which consists of dividing a node into sub-nodes repeatedly choosing always the best option.

##### Cellular/Mobile Networks

This technology is based on cellular base-stations, which provide a long coverage area. Here, there are four standards used in the current studies 2G, 3G, 4G, and in last years 5G [74,96,108]. The improvements of each standard along the years permit to achieve better position estimation. Thus, with 5G technology is expected to have a more accurate position estimation than previous technologies such as 3G or 4G.

In the analysed studies, the authors combined cellular, Wi-Fi, BLE, and Radio Frequency Identifier (RFID) in order to provide a more accurate solution. For instance, Santa et al. [73] developed an indoor-outdoor positioning system which supports five technologies Wi-Fi, BLE, cellular, Near-field Communication (NFC), and Global Positioning System (GPS), in this case the position estimation is done in MEC nodes. Despite combining five technologies the positioning error was around 4.61 m.

In the same fashion, Kulshrestha et al. [96] developed a real-time surveillance system capable of tracking individuals mobile devices. In the proposed system, the authors combined three technologies, GPS, cellular, and Wi-Fi, to provide a robust solution for indoor and outdoor environments. This system was tested in a real environment—a festival that took place in India. The authors were able detected crowded places, average visiting frequency, number of individuals with a high level of accuracy. Additionally, the authors used *k*-NN to detect the location of outliers, NP-Hardness and a novel algorithm, namely Latest Locations Retrieval (LLTR), to select the number of personal sensing units (PSUs) needed to find the last location of a person.

##### IEEE 802.15.4—Zigbee

Zigbee is a standard for Wireless Sensor Network, which is highly scalable and has an ultra-low power consumption. In the last release, Zigbee supports Centralized and Distributed Security Network. This technology has gained widespread popularity in smart environments due to its low cost and power efficiency [22].

Generally, multiple Zigbee nodes are deployed in an environment as a network in order to determine the device position. For instance, Li et al. [62] deployed Lbeacons which contain Bluetooth and Zigbee components. In this kind of implementations, one node is the coordinator, which manages the Zigbee network connection, and is also responsible for collecting data from the nodes. Chen and Huang [13] used a network of Zigbee as the main indoor positioning technology.

##### Radio Frequency Identifier (RFID)

RFID is a RF-based technology which operates in low frequency (30 kHz to 500 kHz), high frequency (10 MHz to 15 MHz), Very High Frequency (VHF), Ultra High Frequency (UHF) (850 MHz to 950 MHz, 2.4 GHz to 2.5 GHz, 5.8 GHz), and microwave frequency [6,109]. In general, a full solution for indoor positioning based on RFID technology consist of three main components, tags to identify the device, which requires a reader called interrogator to know the tags information, and RFID antennas [110]. RFIDs are classified into three types, active, passive, and semi-passive. Active RFID uses a local power supply that keeps powered the microchip and the antenna. Moreover, active RFID transmits its ID periodically. Passive RFID can operate without a power supply, and the semi-passive uses a tiny battery to power the micro chip [109].

As example of passive RFID system can be found in Fang et al. [110]. The authors deployed a full passive RFID indoor localisation solution. In this system, the RFID antenna reads the tags data, and the reader transmits the collected information to the host computer. The proposed system achieved an average accuracy rate of 88.1%. In the same fashion, Datt et al. [111] used this technology to provide navigation services for visually impaired people. In this case, it is important an accurate solution that provides information about the environment in real time, such as obstacles or people around.

#### 4.3.2. Magnetic Field

Magnetic fields are used for indoor positioning and navigation systems as they proved to be stable over the time and the IPS does not require any additional infrastructure deployed in the environment. Nowadays, current mobile devices are already equipped with sensors to measure alterations in the magnetic field. Moreover, this technology has been used in several indoor positioning solutions, including commercial solutions as mentioned in  Brena et al. [22]. Similar to Wi-Fi or BLE, fingerprinting technique is also used with this technology because each scenario presents different levels of intensity (i.e., the ferromagnetic materials used in the buildings and other material distributed in the environments generate unique magnetic conditions).

For example, Liu et al. [112] developed a geomagnetism-based indoor navigation system, which combines fingerprinting technique, particular filter (PF), and Dynamic time warping (DTW). First, the geomagnetic signals are stored in a database in order to form the fingerprint dataset. In the online phase is performed the matching stage where the incoming signal is compared with the stored signal, to estimate the user position by using PF algorithm or DTW algorithm and *k*-NN for path selection.

#### 4.3.3. Inertial Technology

This technology uses inertial sensors such as gyroscope, acceleromenter, and magnetometer to estimate the user position. The assembly of these inertial sensors is called IMU. Inertial technology tends to accumulate errors that are proportional to the travelled distance; the greater the distance, the greater the error [22]. For this reason, this technology is commonly combined with other technologies like Zigbee, Wi-Fi, and/or BLE [52,55,113], among others. For example, in Carrera V. et al. [76], UWB and inertial sensors are combined to provide a highly accurate tracking system. As a result, the authors achieved an average tracking error of 0.44 m.

In Nikolovski et al. [83], the authors use the accelerometer and gyroscope for falling detection. The analysed system combined different techniques to reduce the noise produce by the environment or by the sensors themselves. Thus, multiple filtering techniques have been used in this article, such as particle filter and KF. As a result, the authors provided a reliable system for Ambient-Assisted living.

Dead Reckoning (DR) is a common technique used with IMU, this technique used the previous position to estimated the current user or device position. When this technique is oriented to pedestrians, it is called Pedestrian Dead Reckoning (PDR),which is frequently used in navigation systems along with other technologies. Thus, with PDR we can estimate the user orientation, detect steps, and estimate the step length. For instance, in [59] were combined two techniques, PDR and image-based technique for localization, navigation, and path finding. As as result, the authors provided a seamless and energy efficient indoor navigation solution.

#### 4.3.4. Computer Vision-Based Technology

This technology is based on images obtained from cameras that are processed in order to detect different objects in the environment [22]. Current indoor navigation platforms use this technology along with AR techniques in order to provide information of the environment [59,114]. Given the significant demand for computing resources, most of the processes are moved from mobile devices to the Cloud [115] as we saw in previous paragraphs.

Computer vision-based technology is also used with wearable devices such as smart glasses. For example, Ref. [11] combined vision and sonar sensors in a wearable belt, which is used to identify object or obstacles detected with the sonar sensor. Once the images are capture by the device, they are uploaded to the Cloud server for image processing. Fuzzy Logic is the main technique used by the authors, this technique helps to determine the object proximity (near, far, medium) and other inferences.

Unlike technologies like Wi-Fi and BLE, with computer vision it is possible to achieve decimetre-level accuracy, for instance, in [115] the authors acquired a localisation error of 60 cm (approx.) in an area of 7.2 m × 5.2 m. Additionally, the researchers tested their system in three different scenarios of different sizes and obtained similar results.

Simultaneous Localization and Mapping (SLAM) is a well-known algorithm used along with computer vision-based technology but is not limited to this technology [9]. SLAM algorithm is used to reconstruct the ground covered by extracting some features from images obtained from computer vision-based technologies (Visual-SLAM) [24]. For instance, Ref. [68] combined ORM-SLAM with EC, splitting the computing load between the mobile device and the EC. Thus, the *tracking* process is done in the mobile device and the *local mapping* and *loop closing* in the Edge. This combination allows optimizing the mobile resources with a minimum increment in the positioning error and the map reconstruction.

#### 4.3.5. Sound-Based Technologies

Sound-based technologies are divided into two main groups ultrasound and audible sound. Ultrasound uses frequencies greater than 20 kHz, which is not perceptible for the human ear. Meanwhile, audible sound operates with frequencies less than 20 kHz [6]. Ultrasound is used to detect object or obstacles in the environment. Due to that its speed is slower than other technologies, the time of flight between the transmitter and receiver can be measured [22].

In [11], the authors used ultrasonic sensor to detect obstacles in indoor environments. In this study, two ultrasonic sensors have been used and integrated using a KF. As we can see in previous paragraphs, this technology can be fused with computer vision-based technology in order to provide navigation assistance for visually impaired persons.

#### 4.3.6. Optical Technologies

Optical technologies have increased their popularity in IPS, given their low power consumption and rapid switched on and off intervals [22]. The most common optical-based technologies are VLC and infrared. VLC is gaining popularity for IPS, as it is not susceptible to electromagnetic interference and it provides high-precision positioning and it is low-cost [12]. However, the ambient light might affect its performance, among other factors. That is why different techniques and standards have been developed to overcome these challenges with VLC. For instance, IEEE 802.15.7 is a standard that defines MAC and physical layer (PHY) layers for short-range optical communications.

Similarly, unmodulated light is used for indoor navigation and localisation using light sensors to measure the light intensity. In this case, the ambient light is sensed through light sensors detecting light intensity variations. For instance, in [116] the authors combine unmodulated luminaries with inertial sensors in order to provide an accurate indoor navigation solution. This study aims to determine the light intensity provided by unmodulated luminaries distributed in the environment. Thus, in this study, the peak of light intensity is associated with a virtual graph to determine the navigation path between luminaries with an accuracy of 90% (approx).

Table 3 summarises the technologies analysed in this section together with their group, range, accuracy, and power consumption. As can be seen in this table, each technology has its pros and cons, for instance, cellular networks have a wide coverage area, but their accuracy is low. Conversely, UWB provides high accuracy, but its coverage is limited to 10–20 m.

### 4.4. Do the Current Platforms Adapt to Different Scenarios? (RQ4)

The indoor scenario is considered a highly complex environment for positioning, localisation and navigation purposes, given the diversity of space distributions (rooms, open areas, corridors), building materials, obstacles distribution, and other mobility restrictions.

Thus, different factors have to be considered prior to the deployment of a particular IPS in multiple environments, such as technologies available, the size of the area, the number of floors, the cost, and the accuracy required, among many other factors. That is why some authors have implemented their solution in new locations based on previous experiences and assumptions. Currently, many of the proposed solutions have been deployed and tested in shopping malls [45], universities [46], libraries [119], residence buildings [16], among other public large areas.

In order to deploy an indoor positioning solution in different environments, we have considered three main aspects related to namely the platform (software), the environment, and the client device, which are addressed in the following paragraphs.

#### 4.4.1. Platform

Section 4.1 addressed the computing paradigms used in the studies analysed. As we can observe, the developed systems were deployed by using different computing paradigms such as the CC, EC and FC. Some of them correspond to well-tested IPS/ILS that have been used by many organizations. For instance, AnyPlace [55] is an open-source indoor navigation platform, which is available online for free, and it can be used in many environments, including industrial ones.

Each indoor positioning platform has differing requirements in terms of software and hardware, and they are specified in their documentation. These requirements are essential to the proper operation of the system. The deployment of these platforms are becoming less complex due to the fact that the authors use Cloud services such as those provided by Amazon, Google, or other companies. For instance, Terán et al. [48] and P. Álvarez and N. Hernández and Fco Javier Fabra and M. Ocana [45] used some services provided by Amazon such as S3 Service, Amazon API Gateway, AWS Lambda, Amazon Dinamo DB, and Amazon Machine Learning.

Additionally, these platforms have to be adaptable to both new and old indoor positioning technologies. Thus, if the indoor positioning platform only supports one technology, it will take more time to adapt it to new scenarios where that specific technology is not deployed or cannot be deployed. Current commercial solutions try to provide a wide range of supported technologies to be used in different environments together with heterogeneous technologies.

#### 4.4.2. Environment

Some of the studies analysed tested their indoor positioning and localisation platforms in a single scenario [48,58,72], other platforms were tested in more than one scenario [55,82,119], including outdoor environments [120]. Thus, the experimental area might differ from a few square meters to several hundreds of square meters in each reviewed article. For example, P. Álvarez and N. Hernández and Fco Javier Fabra and M. Ocaña [45] tested their Cloud-based location and tracking system in a scenario of 562,000 m^2^,   Liu et al. [116] implemented their indoor navigation platform in three environments, a supermarket of 1000 m^2^, a shopping mall of 20,000 m^2^ and an office building of 800 m^2^. Sujin et al. [121] evaluated their system in an area of 144 m^2^.

The diversity of indoor environments has led some researchers to carry out a previous environment survey [56] to train their applications with ML algorithms and fingerprinting technique (or similar methods) [7,56]. For instance, Noreikis et al. [59] attempted to take a video from the venue to form 3D point Cloud. It is required to provide localisation and navigation services in a new environment. In contrast, the navigation service developed by Konstantopoulos et al. [42] does not require a training stage to be used in a new scenario, it only needs the POIs, the predefined routes and the information provided by the beacons (RSS values) to provide the navigation service.

#### 4.4.3. Client

In this case, the client-side is linked to the mobile devices. In the analysed studies, the researchers used smartphones, wearable, and IoT devices were used to collect information about the environment and display the information related to the location, position, route, among others. Thus, the authors have developed applications for those devices which allow the interaction between the user and different service. Therefore, these mobile applications process, exchange, and consume services from the IPS/ILS and third-party services such as Google maps. The data can be exchanged through APIs [40,50,51,52,54,55,77,78,79,114,120], web services [48], or simple http(s) request [102], allow an straightforward integration.

Since these applications are easy to install in many of the current Operating System (OS) (Android, iOS, Windows), they can be used in any environment with their own indoor positioning platform after minor changes. For example, Facchinetti et al. [54] developed a mobile android application, namely “IPSOS assistant”, which is used to guide people in case of emergency such as users’ injuries, and paths to emergency exits, among others. This application consumes some Google services (e.g., maps, Google Cloud Messaging (GCM)) and the services provided by the SOS server. The communication between the application and the server is done through a REST API service. Thus, most of the configuration is done on the server-side. Similarly, Raspopoulos et al. [79] developed a mobile app that interacts with the Cloud platform using an API.

### 4.5. What Improvements Were Done in Similar Studies (RQ5)

Each research study is focused on improving and solving current gaps in a specific research field, in this case in Cloud-based indoor positioning platforms. This research field combines different areas such as Cloud Computing (CC), indoor and outdoor positioning, signal processing, and software architecture. It should also be noted that some of the analysed articles are focused on more than one objective. That is why we can find many goals linked to each of the mentioned areas. To address this research question, we have created a framework with sixteen items summering each study’s main improvements.

It is important to highlight that we do not use position and localisation as synonyms, as those terms have different meanings. However, some of the articles did use those terms interchangeably. Sithole and Zlatanova [122] define four terms: position, location, place, and area, where the position is an exact point in the space denoted by a coordinate (x,y,z or latitude and longitude), whereas location is a small physical space, for instance, *room A-123*, *Geotec Laboratory*, etc. As we can see, these two terms express two different meanings, and they should be used appropriately in the articles.

Efficient Computation [45,48,52,61,68,74,85,94,102,115]: It consists of improving the methods or algorithms used in mobile devices and the Cloud in order to decrease the use of computational resources. To reduce the computational load in mobile devices, the authors offload specific processes to the Cloud or other computational paradigms (see Section 4.1). Moreover, researchers have proposed some optimizations to traditional algorithms and databases in order to improve their efficiency and time response.Interoperability [54,78,79,95]: It is the capability to interact with other systems, platforms, or devices through its interfaces. Thus, they can exchange information simultaneously, allowing them to integrate with each other and provide synchronous communication. This is especially valuable in light of the heterogeneity of the deployed IPS, and the need for position and localisation services in other areas such as healthcare systems, i.e., if an ILS only shares the estimated position and does not provide interfaces to share raw data, these raw data cannot be integrated into a sensor fusion approach.Position [55,56,58,74,85,123]: The articles studied proposed different technologies, techniques, and methods to reduce the error in the position estimation (see Section 4.3). Additionally, the use of computing paradigms (e.g., CC, FC, MCC, EC) have been used in some articles to support the positioning process.Usability [8,13,39,40,43,77,79,80,83,96,97,106,111,124]: It is linked to the user experience providing a platform easy to use that satisfy the user’s requirements. For instance, Yeh et al. [39] developed a Cloud platform for parking services (e.g., search parking places, reservation, navigation), providing a useful and efficient system to end-users which satisfies the need for parking systems. Additionally, some of the applications or frameworks are oriented to developers or users in general who have limited knowledge of positioning systems and programming, allowing fast development of a new application.Localisation [7,10,16,42,58,72,75,76,82,107,110,114,121,125,126]: Similar to position, localisation aims to provide better localisation accuracy by combining different techniques, technologies, and algorithms. In the current studies, localisation techniques have been used to locate people in different environments such as shopping malls, universities, hotels, among others.Cost [45,67,88,94,105]: The cost is one of the prime considerations when researchers and companies develop their indoor positioning platforms. That is why technologies like Wi-Fi and BLE have been chosen, despite their poor accuracy compared to, for instance, UWB. Moreover, the use of Cloud Computing (CC) offers pay-as-you-go, enabling users to pay only for the services and resources procured.Navigation [11,46,55,59,69,99,111,112,120,127]: Many of the current studies are focused on improving or providing navigation service. For instance, to provide navigation services for shopping malls or select the best route to emergency exits. The navigation service is also used to choose the least congested route to a particular place.Scalability [62,110,120,123]: It is the ability of increasing the capacity of the platform in terms of operational area, position technologies supported or number of concurrent users without inhibiting the performance. Only four works have analysed/considered this dimension in their platforms.Low latency [64,106,111]: This is a critical point in time-sensitive networks, and it is related to the delay in the data transmission. It is the time that it takes a message to go from the source to the target. It is specially required for real-time communications. Thus, in order to provide real-time indoor positioning/localisation/navigation applications, the authors use different technologies, techniques and computing paradigms (see Section 4.1 and Section 4.3) in order to reduce the latency, for instance, FC is used for facing the latency problems caused by the large number of connections to the IPS.Energy efficiency [58,59,103,116,128]: Several measures have been taken to reduce the energy consumed while performing a task. However, the main method used in the current studies is to offload certain processes from the mobile device to computing paradigms. This allows the use of IPS in low-profile devices such as IoT and wearable devices.Reliability [8,110]: To provide reliable positioning and localisation information in a variety of environments with a minimum of errors, providing a high-quality service. However, providing reliable systems is not easy given the complexity of IPS/ILS. For instance, the authors of [110] implemented their system at a building construction site in three scenarios in order to test the accuracy, latency, and system reliability, obtaining a precision of 85% and accuracy of 88%, approximately. Ref. [8] developed a reliable localization web platform providing remote access to numerous users.Tracking: It determines the current user position in real-time with minimum delays. In order to provide tracking services, the authors use certain algorithms and technologies. For example, Sujin et al. [121] used a stochastic model, namely the Markov model, which is used for device tracking.Evaluation: It is one of the important aspect for indoor positioning platforms in order to determine the performance and if it fulfils all the technical and user requirements. This evaluation could be carried out in simulated environments or real environments following a specific standard similar to the platform developed by  Haute et al. [43].Privacy [70]: Given that some of the information collected to train positioning and localisation models might contain sensitive data, the authors provide some mechanism to protect the privacy of the user information during the process in any of the computing paradigms. For instance, Zhang et al. [70] applied differential privacy to Edge-based IPS. This research aims to protect user information when it is used to train positioning localisation models in the EC.Security [63,71,91,119,129]: Various techniques, protocols, or devices have been developed over the year to protect user information. Thus, protocols like HTTPS are widely used to provide safe data transfer. Moreover, other mechanisms have been adopted to determine anomalies during the data collection and verify security issues in indoor positioning platforms.

### 4.6. How Is the Standardization Aspect Focused on Different Platforms? (RQ6)

The standardisation provides a common language to interact between different systems, which is highly relevant when we have to integrate heterogeneous systems. Moreover, it enhances the interoperability between devices and systems of multiple vendors. Nevertheless, not all the platforms for positioning and localisation follow the standards established by the standardisation entities. Nowadays, different standard organisations provide guidelines for products, services, education, process, and other fields of interest.

This research question addresses four components of indoor positioning/localisation platforms, which are listed below.

#### 4.6.1. Maps

There are multiple standards for indoor and outdoor maps which establish a set of rules and a common framework, allowing for the exchange of spatial information under the same “language” or format. It includes coordinate reference systems, indoor and outdoor space representation, feature representation, map visualization, among others [130]. Open Geospatial Consortium (OGC) have provided different standards with the aim of making geospatial information accessible for everyone, allowing the interoperability between geographical systems under technical guidelines. As part of OGC standards we have the IndoorGML, which is focused on Extensible Markup Language (XML) schemes for indoor spatial information. Thus, we can represent indoor topographic space, sensor spaces, connection points through XML tags, for instance, *<xs:element name="Geometry3D" type="gml:SolidPropertyType"/>*. In similar a way, Indoor OpenStreetMap provides a complete framework for modelling indoor maps which includes buildings, POIs, connections, indoor elements, among others. For example, Indoor OpenStreetMap uses the following tags for representing indoor environments, *door=yes/hinged/sliding/no*, *capacity=**, *entrance=yes*, etc. Additionally, some authors used Building Information Modeling (BIM) [106,110] tools to model and manage building information enabling easy collaboration and sharing data about buildings and civil engineering works. BIM is under the International Organization for Standardization (ISO) 19650.

Some of the studies analysed take into account or use the current standards. Ref. [77] developed a platform for managing indoor and outdoor spaces and geolocating. The authors followed the OGC standards and the OpenGIS specifications to develop their platforms. They also specified the geodetic system supported by their platform (GGRS87 or EGSA87 and WGS84 [95]). In most of the cases, third-party maps are used in their applications, for instance, Google Maps [40,54,80,82,99,111,120], or custom maps [46,50,67,74,88,95,119,121].

#### 4.6.2. Position Technologies

Section 4.3 provided a review on the position technologies used on the research studies. Most of them are under specific standards, for instance, Wi-Fi under the family standard IEEE 802.11, Bluetooth standard IEEE 802.15.1, UWB built upon the standard IEEE 802.15.4z, Zigbee built upon the standard IEEE 802.15.4, RFID has several standards including ISO 18000 and EPCglobal.

#### 4.6.3. Evaluation Methods

In order to evaluate IPS/ILS, the ISO provided and standard namely ISO/IEC 18305:2016 Information technology—real-time locating systems—Test and evaluation of localisation and tracking systems [44]. This standard contains different metrics for Localization and Tracking System (LTS), privacy and security considerations, performance metrics, which was thought of as guideline to standardize IPS and has been largely discussed by the indoor positioning community [131,132]. This standard mentions the importance of evaluation in IPS, to know if the platform fulfils all the user’s requirements. Upon this standard [43] developed a platform devoted to comparing test indoor positioning solutions.

#### 4.6.4. Software Architecture

Software architecture is related to the software design, structure, its components and the relation between them. It has to be considered as the foundation of any software development, given that it directly affects the quality and reliability of the software. Current studies have considered some of of the software architecture, such as methodologies, software architecture patterns (e.g., Model–view–controller (MVC) and Model–view–viewmodel (MVVM)), Service Oriented Architecture (SOA), monolithic, Microservice Architecture (MSA) and Cloud native architecture. The IEEE 42020-2019—ISO/IEC/IEEE International Standard—Software, systems and enterprise—Architecture processes [133] provides a full description of software architecture, definitions, process, implementation, and other relevant information.

As part of the software architecture, the authors used different software design architecture patterns, but in some cases, it is not mentioned or described in the research study. For instance, SOA was used in [42,54,78], this architectural pattern is oriented to develop services that work together to automate processes. In such a way, each service has a specific function or task in the business process, reducing the software’s complexity. Mpeis et al. [55] used the MSA patter to develop their indoor navigation platform. MSA use small independent microservices, which have their interfaces to communicate with other services over standard lightweight protocols (e.g., MQTT, XMPP). Generally, this kind of architecture is used for large and complex solutions. For instance, Netflix, Uber, and Amazon applications are developed using this architectural pattern [134]. Cloud-native architecture is used by [40,46,48,59,97] to be deployed in the Cloud. Thus, these indoor positioning/localisation/navigation platforms take advantage of Cloud computing or similar computing paradigms.

## 5. Discussion of the State of the Art

The six research questions addressed systematically throughout Section 4, using the methodical procedure laid out in Section 3. This section summarizes the systematic review’s main findings, discussing current challenges and future trends.

### 5.1. Computing Paradigms and Improvements (RQ1 and RQ5)

Section 4.1 reviewed all of the computing paradigms used in the analysed research works. Here, we can distinguish six computing paradigms used in indoor positioning/localisation platforms, being CC and MCC the most popular ones, with 35 and 24 articles, respectively.

Moreover, the CC has been used in combination with other computing paradigms like EC, FC, and MEC (see Figure 4). As discussed in previous sections, positioning and localisation services are increasingly utilised in different areas and environments thus demanding ever greater computational resources. Therefore, robust equipment is required to bring a high-quality service.

Despite the CC advantages, there are some issues regarding latency and security & privacy. These issues led some researchers to employ alternative computing paradigms. For instance, FC was used in 12 research studies (see Figure 5) in order to allow massive device connectivity providing a fast response to the end-user. Similarly, FC is used to offload some computational processes from the mobile devices to this paradigm. As a complement of FC we have MIST computing, which extends the FC capabilities as mentioned in Section 4.1.4. In the same fashion, EC and MEC provide lower latency than CC, bringing increased storage and computational capabilities.

Given that some of the algorithms used to estimate user position or localisation are extremely demanding of computational resources, it is almost impossible to run them in low-profile devices such as wearable devices (e.g., smartwatches, smart glasses). The greater the complexity of the algorithm, the more computational resources are required. Thus, offloading these heavy processes to the computing paradigms reduce the computational load in mobile devices. Therefore, the battery life may be significantly affected when indoor positioning applications are running on them, because data collection (sensing) and communications are still needed.

In the last seven years, the use of computing paradigms has increased in this research area. However, the state of the art of computing paradigms has been employed less in comparison to CC and MCC. With the massive device connectivity of IoT and mobile devices, it is necessary to use these computing paradigms, which are capable of supporting hundreds and thousands of devices connected to them, providing low latency communications.

Figure 5 shows the goals achieved in each analysed study divided by computing architectures used by the authors. Most of the analysed works deal with improving *localization*/*positioning* accuracy (17 + 8, 25 papers total), *efficient computation* (16), *navigation* (12), and *usability* (17). On the other hand, relevant goals as cost, evaluation, functionality, latency, privacy, reliability, scalability, security or tracking appear in 7 or less papers.

From this figure, we can also see that the computing paradigms are somehow correlated to the specific goals. For instance, the analysed research studies that use CC and MCC are chiefly oriented towards *usability*. FC and MEC-based are focused on providing *efficient computation*, and EC-based concentrates on providing an accurate *localisation*.

In the current research studies, the authors attempt to achieve more than one objective in some cases, such as providing an accurate and low-cost indoor positioning solution or providing reliable and scalable indoor positioning platforms. Although some data are processed in the Cloud or other computing paradigms, security and privacy are not the primary concern. As shown in Figure 5, security and privacy are under-researched topics, addressed in only 7 and 4 articles respectively.

### 5.2. Network Protocols RQ2

Network protocols are a fundamental part of Cloud-based indoor positioning platforms, allowing communication between devices and systems, and enabling secure and fast communications. Section 4.2 analysed the protocols implemented in the literature reviewed, separating the network protocols into four categories: communication protocols, security, IoT and other protocols (see Figure 6 (left)). As discussed, the authors may choose the protocols according to the system requirements. For instance, if the platform is oriented to wearable and IoT devices, it is indispensable to use lightweight protocols such as MQTT, UDP and XMPP. Otherwise, the traditional protocols can prove more than suitable for IPS/ILS.

To provide security and privacy in communication between the client and the positioning server, the authors use SSL/TLS protocols. However, it can be time- and energy-consuming given the processes carried out by these protocols (e.g., TLS handshake), which could not be suitable for some IoT devices. Although some security and privacy algorithms have been proposed in the literature, they are not standards protocols, and were designed for computing paradigms such as the EC and not for power-constrained devices.

Selecting the most appropriate network protocol is a critical factor in the provision of reliable and efficient indoor positioning solutions, which fulfil all the user requirements, including security and privacy.

### 5.3. Indoor Positioning Technologies (RQ3)

A wide range of indoor positioning technologies have been used and tested in the analysed literature (see Figure 7), being the predominant RF-based technologies such as Wi-Fi, which was used in approximately 67% of the reviewed studies. Although Wi-Fi is not the best solution in terms of positioning accuracy, it is the most suitable given that Wi-Fi Access Points (APs)/routers are already deployed almost everywhere for communication purposes. In the second place, we have Bluetooth-based indoor positioning solutions (31 articles), which provide a lower error than Wi-Fi, and their cost is relatively low. Moreover, in the case of BLE, it also provides low power consumption being used by many power-constrained devices.

However, technologies like UWB and camera provide a lower positioning error than BLE and Wi-Fi (see Section 4.3). These technologies have been less used in the reviewed studies. In the case of UWB, it is necessary for some extra hardware to be deployed in the environment in order to estimate the user position. Moreover, not all the user devices support UWB technology, becoming a limitation for its implementation. However, in the case of camera-based, most mobile phones are already equipped with high-resolution cameras, but these camera-based solutions require more computational resources as image processing has attached a huge computational burden.

Hybrid solutions (i.e., sensor fusion combining multiple technologies such as RF, sound, inertial sensors, etc.) has become a hot topic in recent years. As we can see in Figure 7, 20% of the analysed studies combined more than one technology to provide a highly accurate solution or provide a wide range of options to the end-user. Commercial and open-source solutions tends to offer indoor navigation and positioning systems which support Wi-Fi, BLE, UWB, and other technologies, for example, AnyPlace [55].

Here we can distinguish that inertial sensors are used to support other technologies like Wi-Fi, BLE, and camera-based, being the third more used technology which appears in 15 articles. However, it tends to accumulate errors that are proportional to the walked distance. That is why some filters (e.g., KF or particle filter) have been applied to reduce the error when this technology is used.

Similar to technologies, the authors of the analysed articles used numerous techniques and positioning algorithms to reduce the positioning error. However, the most used positioning technique is still fingerprinting, as we can observe in Table A1 included in the Appendix A. Out of algorithms used to estimate the user or device position, many are based on ML algorithms such as SVM, *k*-NN, and LSTM. Here, refs. [48,56,94,112].

### 5.4. Cloud-Based Indoor Positioning Platforms—Scenarios (RQ4)

As we mentioned in Section 4.4, the indoor environment is one of the most complex scenarios for positioning, localisation, and navigation systems. This leads to large positioning errors in the position estimation. That is why multiple technologies, techniques, and methods have been developed in order to tackle positioning indoors. The current solutions have to be adaptable and flexible to be used in heterogeneous scenarios.

To provide adaptable indoor positioning/localisation solutions, we have to consider three essential aspects. The first one is the system (platform), which should be easy to deploy in any Cloud or on-premise environment with minor modifications. Moreover, these platforms have to support the most common indoor positioning technologies and extend their capabilities to new technologies and services. As we can see in previous sections, many of these platforms offer Cloud-based positioning services such as navigation and positioning services. In this case, it is not necessary to deploy them in other Cloud platforms or locally; just consume their services, for instance, under the model pay-as-you-go.

The second aspect is focused on GNSS-denied environments. In Section 4.4, we can see that most of the platforms have been tested in heterogeneous environments, including universities, hospitals, malls, offices, among others. These environments have different characteristic and different necessities in terms of accuracy, time-response, among others. Thus, GNSS-denied scenarios are widely studied previous the deployment of any indoor positioning technology.

The last point is related to the client-side if the indoor positioning applications can run in any OS (e.g., Android, iOS, windows). Currently, most of the applications developed for indoor positioning can be installed in different devices and OS. These devices are used to collect indoor information (e.g., RSS values, images, sensor data, etc.) and show the position or localisation results to the user. Additionally, some platforms also offer solutions for wearable and IoT devices, avoiding additional developments from the client side.

### 5.5. Standardization (RQ6)

It is of paramount importance to develop systems following international standards (ISO, IEEE, etc.). This is the way to obtain a straightforward integration between systems and components, i.e., reaching interoperable IPS/ILS able to collaborate under an advanced sensor fusion umbrella. It has also been proven that following standards enables a faster technological innovation as well as a higher quality in provided services and products.

In Section 4.6, we analysed how the standardization aspect is addressed in the reviewed articles. Given that indoor positioning platforms are formed of different components, we analysed four components related to maps (indoor and outdoor), position technologies, evaluations methods, and software architecture. In spite of the different standards for indoor maps (IndoorGML, Indoor OpenStreetMap, Indoor Mapping Data Format) and the efforts done by the community to promote them [130], the current studies do not mention the use of any of them. However, standards like OGC and BIM have been used in some of the reviewed articles.

Similarly, the use of standards to test and evaluate localisation systems has used or mentioned in only one article. In spite of standards like ISO/IEC 18305:2016 [44] provided a full guideline and metric to evaluate the performance of indoor localisation and tracking systems. Moreover, it provides some considerations about security & privacy, indoor scenarios, reporting requirements, and others.

In the case of software architecture standards, the analysed articles are more focused on software architecture patterns. Here we can identify three design patterns SOA, MSA, and Cloud-native architecture (see Figure 6). Each of these design patterns has advantages and disadvantages, and the developers have to select the best approach for a given indoor positioning solution. Given that most of the platforms are deployed in the Cloud, the developers should take advantage of all the benefits it provides.

Although the use of standards has to be the cornerstone of any application or system, including IPS/ILS, not all the analysed articles provide information about which standard was used in their applications.

### 5.6. Current Challenges

In this section will be described current challenges obtained from the analysis from the primary studies. Some challenges are related to computing paradigms, scenarios, and standardization.

#### 5.6.1. Challenges Related to Computing Paradigms

Security and privacy are a crucial concern in the computing paradigms CC, EC, FC, MCC, and MIST computing. Given that these computing paradigms support massive device connectivity, the risk of security and privacy issues increases. However, the number of studies that addressed this topic were approximately 13%, and around 6% of it was oriented to cc and MCC. The remaining 7% was focused to ec and FC. However, there are many open questions regarding this hot topic and Cloud-based indoor positioning platforms. For instances, how to efficiently detect anomalies during data collection. How to ensure the sensitive data collection for positioning and localisation purposes, among others.

#### 5.6.2. Challenges Related to Software

Software architecture design is essential in any application or system. This will determine the robustness, scalability, fault tolerance, usability, interoperability, and other characteristics of high-quality systems or platforms. In the current analysed studies, we can observe that the authors are focused on many of these characteristics of design software. However, some characteristics are still missed, for instance, fault tolerance which is related to the time response and service recovery in case of failure in one or more components without stopping the service.

Additionally, we can see that computing paradigms are becoming more and more used in indoor positioning and localisation system, given the high capabilities. This leads us to think about the flexibility of the current indoor positioning/localisation systems to be deployed in one or many computing paradigms, including the capability of distributing specific modules according to the computing layer used. This would involve a redesign of many platforms to be adapted to new computing paradigms.

#### 5.6.3. Challenges Related to Standardization

Although there are some standards available for indoor positioning localisation systems (e.g., ISO/IEC 18305:2016 [44] and IndoorGML). It is not easy to find out information about them in the analysed studies. The standardization of indoor position and localisation systems is one of the main concerns in both academy and industry. That is why some standards have been developed, which lead to interoperable and well-tested systems under rigorous guidelines.

However, standardization is not only related to mapping, communication technologies and evaluation standards. Here, it is important to consider the standards used for software development and interoperability, such as using the standard ISO/IEC/IEEE 42010 Systems and software engineering—Architecture description. In such a way, it will permit the fast development of reliable and quality indoor positioning and localisation systems.

### 5.7. Future Trends

Both applications based in ILS/IPS and Cloud-based solutions are being used more and more in the last few years. For instance, during the COVID-19 pandemic have been developed numerous frameworks and applications for contact tracing. These applications aim to protect people by informing them if they have been in contact with people with COVID-19 or exposed to this virus. Thus, contact tracing application determines the proximity between people in indoor and outdoor environments by using Bluetooth technology and mobile devices.

Additionally, LBS are becoming more used in many applications and research fields, including e-health, sports activities, rescue systems, social networking, autonomous navigation, among others. As noted earlier, this leads to having million of devices connected and consuming positioning, localisation, and navigation services. Thus, the current trend is to use computing paradigms to give support indoor positioning and localisation systems. Moreover, the use of hybrid localisation technologies (sensor fusion) is evident in current research as they provide better positioning results.

Despite of most of the positioning and localisation services being used in mobile devices (e.g., smartphones, tablets), there is also an increasing trend towards its use in wearable and IoT devices. However, the developed applications for these devices have to be as efficient as possible to consume less power and computational resources.

## 6. Threats to Validity

This section is devoted to discussing the threats to validate the current empirical systematic review. Thus, we have considered the following aspect to validate.

### 6.1. Methodology Selection

The PRISMA model was selected to report this review, as it is a well-defined and systematic procedure which provides a replicable and reproducible review. The findings of the reviewed literature are presented together with a discussion of the current state of Cloud-based indoor positioning platforms, challenges, and future trends. The full list of studies selected is available to be evaluated by any person interested in this review and its results.

### 6.2. Primarily Studies Selection

Following the selection of the methodology, we defined a research query to run through two well-known search engines *Web of Science* and *SCOPUS*. By using these research engines, we avoided the danger of discarding indexed publications of different journals such as IEEE, ACM, and Springer, among others, minimizing the risk of obtaining different results. We also defined a general research query to include all relevant results. As a result, we obtained several records for its further process and revision. Section 3 describes the whole process to make the results replicable.

### 6.3. Selection of Studies

In order to select the research studies for this systematic review, we proceeded with the manual revision of titles and abstracts, tagging them with ACCEPTED and REJECTED labels. This selection was made on the basis of the inclusion and exclusion criteria established in Section 3. there was a risk of relevant articles being excluded due to the inclusion criteria and the quality evaluation in Section 3, stage 3 and 4. However, to avoid validation issues, we followed the guidelines established in the PRISMA model.

Our review only covered articles from 2015 to 2021 which related to Cloud-based indoor positioning systems and other computing paradigms which differ from mobile computing. Additionally, we selected only those articles related to mobile devices including wearable and IoT devices. We discarded robots and unmanned autonomous vehicle and other not connected with the research topic. We believe that the risk of an external validation threat is low under these conditions.

### 6.4. Replicability

This work can be easily replicated by following the steps described in Section 3. However, the number of primary records may vary due to the potential for publication of further research papers on the topic. For instance, papers written in 2021 may be published in the early months of 2022.

## 7. Conclusions and Future Directions

This paper has presented a systematic review of the current studies relating to Cloud-based indoor positioning and localisation platforms and similar computing paradigms. We have analysed the varying components which form these platforms, such as computing paradigms, technologies, network protocols, and standards and scenarios. We used the PRISMA model as the basis of our review in order to provide a replicable work and report studies’ main findings. Here, we have defined different research questions that had been answered along with the article in various sections.

This systematic revie, has demonstrated a growing trend in the use of computing paradigms in IPS/ILS from 2015 to 2021. These computing paradigms provide ubiquitous computing, low latency, high computational and storage resources, and have consequently proved a hot topic in many research fields. This was the primary factor in our decision to analyse the use of multi computing paradigms (CC, FC, EC, MEC) within indoor positioning and localisation systems.

Additionally, we have observed that using different technologies, methods, and algorithms, and combinations of these can help us acquire reliable position accuracy and reduce computation time. Nevertheless, there is not a specific technology for indoors, prevailing Wi-Fi over the other technologies such as UWB, BLE, Zigbee, etc. However, Wi-Fi was not designed for positioning purposes.

It is also essential to highlight the use of standards (ISO/IEC 18305, IndoorGML, etc.), including terminology, error metrics, data format, environment description, and other information described in the standards. This will reduce the integration time with other systems, but it will also improve the positioning systems.

We can conclude that it is necessary to develop modular indoor positioning and localisation platform which fulfil all the standards in each of its components. Moreover, these platforms should be capable of adapting to different indoor positioning technologies and scenarios with minimal effort from the client side.

## Figures and Tables

**Figure 1 sensors-22-00110-f001:**
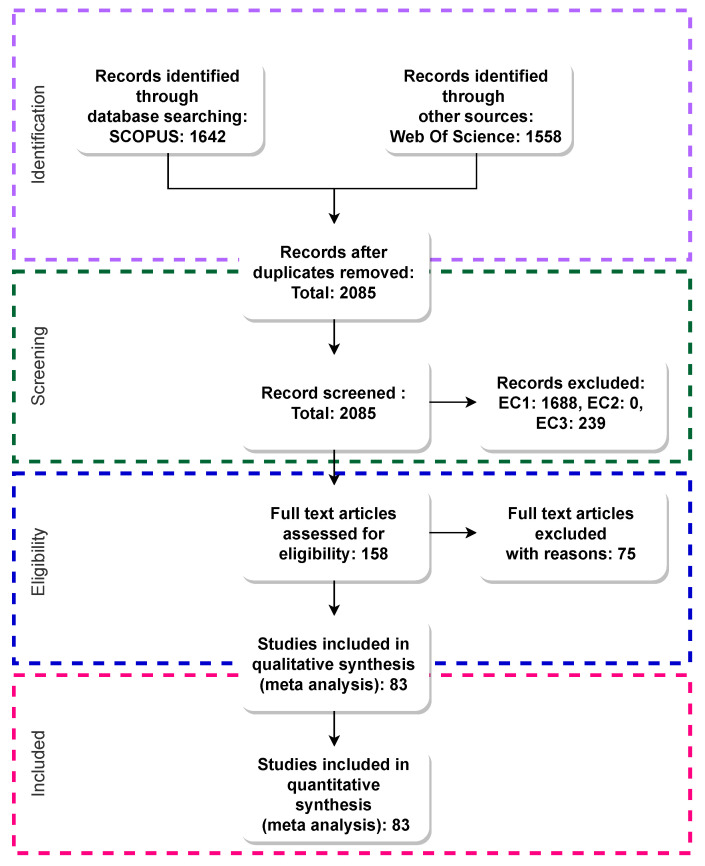
PRISMA Flow Diagram.

**Figure 2 sensors-22-00110-f002:**
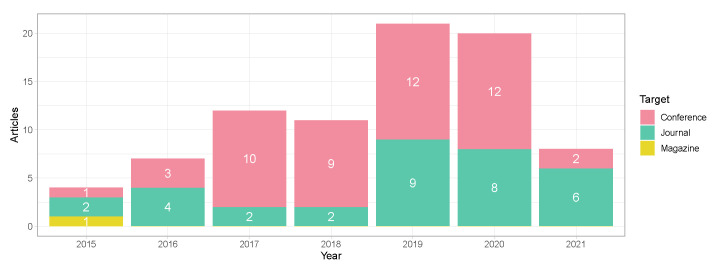
Number of studies related to Cloud-based indoor positioning per year.

**Figure 3 sensors-22-00110-f003:**
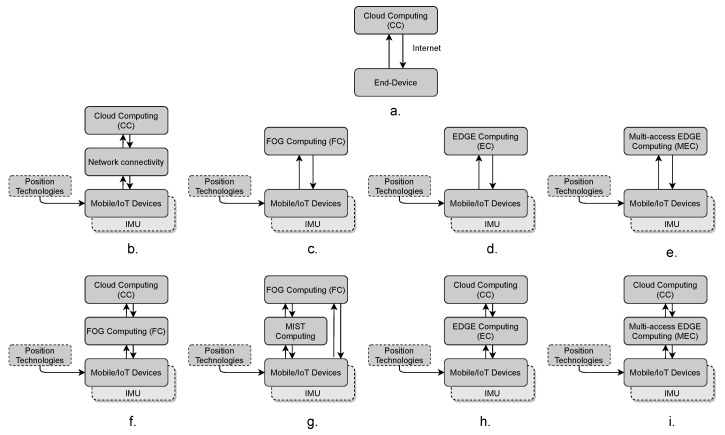
Current Cloud-based IPS/ILS architectures: (**a**) Cloud Computing (CC), (**b**) Mobile Cloud Computing (MCC), (**c**) Fog Computing (FC), (**d**) Edge Computing (EC), (**e**) Multi-access Edge Computing (MEC), (**f**) Mobile-Fog-Cloud, (**g**) Mobile-Mist-Fog, (**h**) Mobile-Edge-Cloud, (**i**) Mobile-MEC-Cloud.

**Figure 4 sensors-22-00110-f004:**
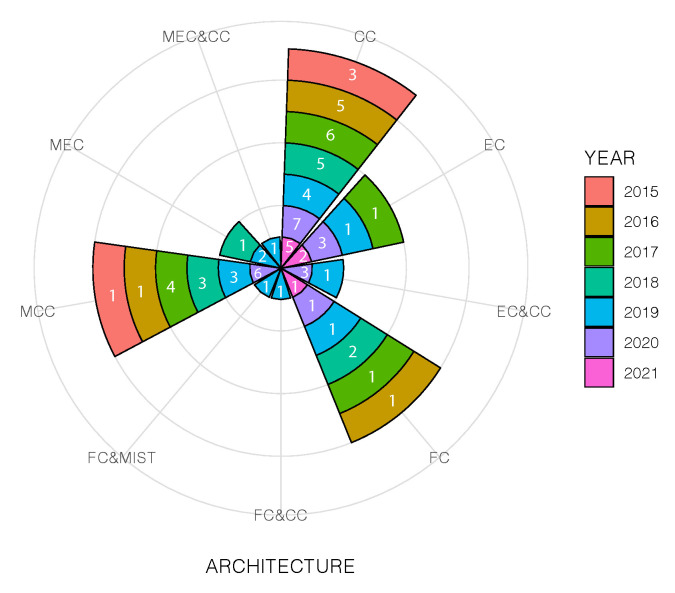
Use of computing paradigms per year.

**Figure 5 sensors-22-00110-f005:**
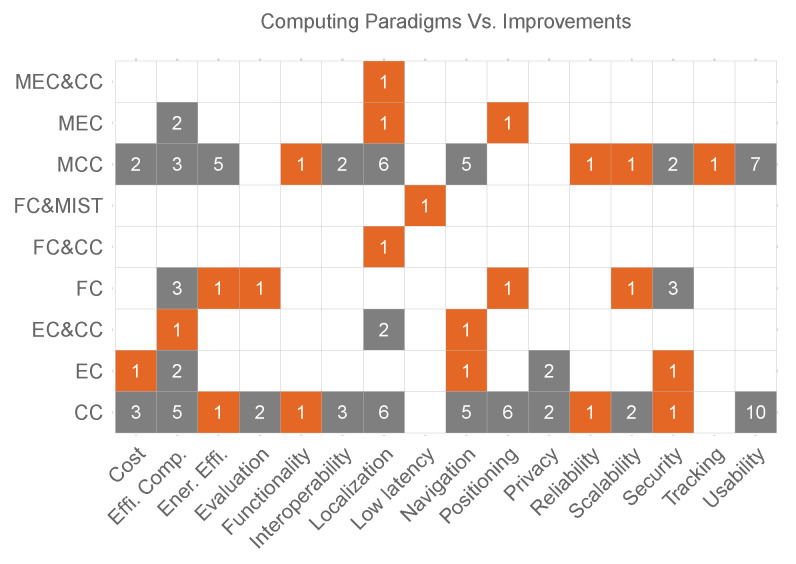
Computing paradigms and studies analysed main goals.

**Figure 6 sensors-22-00110-f006:**
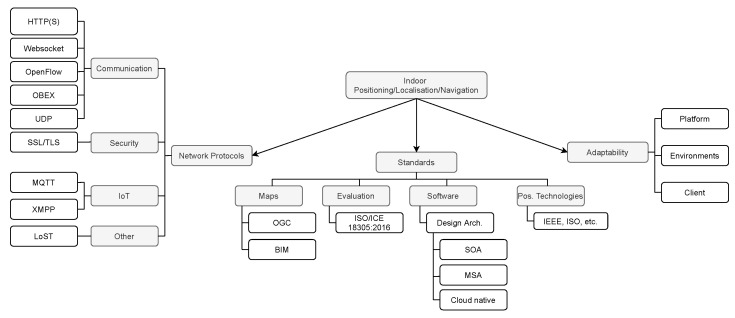
Network protocols, standards and adaptability to new environments.

**Figure 7 sensors-22-00110-f007:**
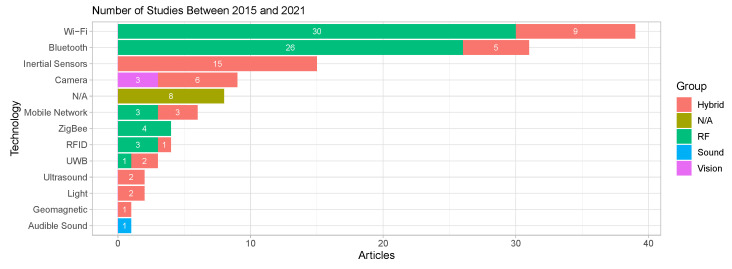
Indoor positioning technologies used in current studies.

**Table 1 sensors-22-00110-t001:** Comparison to other reviews and surveys.

		Approach							
Article	Year	Applications	Technologies	Techniques	Methods	Cloud-Based	Device-Based	Standards	Protocols
[22]	2017	✗	✓	✓	✓	✗	✓	✗	✗
[27]	2018	✓	✓	✓	✓	✵	✓	✗	✗
[25]	2018	✓	✓	✗	✗	✗	✓	✗	✗
[23]	2018	✗	✓	✓	✗	✗	✓	✗	✗
[5]	2019	✓	✓	✓	✓	✵	✓	✗	✗
[9]	2019	✗	✓	✓	✓	✗	✓	✗	✗
[6]	2019	✓	✓	✓	✓	✵	✓	✗	✓
[30]	2019	✓	✓	✓	✓	✗	✓	✗	✗
[28]	2019	✗	✗	✓	✓	✗	✓	✗	✗
[24]	2020	✓	✓	✓	✓	✗	✓	✗	✗
[21]	2020	✓	✓	✓	✓	✗	✓	✗	✗
[26]	2020	✓	✓	✓	✓	✗	✗	✗	✗
[29]	2021	✓	✓	✓	✓	✗	✓	✗	✗
our review		✓	✓	✓	✓	✓	✓	✓	✓

✵ There is no cloud or other computing paradigm analysis, but they are mentioned in the survey or review for their relation with the IoT.

**Table 2 sensors-22-00110-t002:** Keywords related to the topic research.

Keyword Infrastructure	Keywords Environment	Keywords System
Cloud Computing	Indoor *	Position *
Edge Computing		Location
Fog Computing		Localisation
MIST Computing		
Platform		

Wildcard asterisk (*) represents any group of characters.

**Table 3 sensors-22-00110-t003:** Features of the Indoor Positioning technologies.

Group	Technology/Feature	Max. Range	Accuracy *	Power Cons.
	Cellular [117]	500 m–80 km ^*a*^	<50 m [9]	Moderate-low
	Wi-Fi [118]	< 100 m ^*b*^	average > 1 m [67,88,103]	Moderate
	Bluetooth [7,58,104]	v2.1–4.0 → 100 m, v5.0 → 400 m	average > 1.5 m [46,58]	Low
RF	UWB [9]	10–20 m	median < 50 cm [9]	Low
	Zigbee [117]	100 m	median < 5 m [9]	Low
	RFID	200 m	median < 3 m [110]	Low
Optical	Light	-	-	Low
Vision	Camera	-	average ≈ 20 cm [68]	High
Sound	Ultrasound [6]	<20 m	median < 10 cm [9]	Low
	Audible Sound	-	-	Low
Inertial sensors	Gyroscope, accelerometer, etc.	-	<5 m [110] ^*c*^	Low
Magnetic Field	-	-	median < 5 m [9]	Low

* Accuracy reported in the analysed studies and surveys. This error can vary in function of the techniques and algorithm used by the authors. ^*a*^ Depends on the standard (2G, 3G, 4G, 5G), ^*b*^ it depends on the variant IEEE 802.11a, IEEE 802.11b, IEEE 802.11g, etc., ^*c*^ increases as a function of the distance walked.

## Data Availability

Not applicable.

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
