# Peer review of "Cloud Platforms for Context-Adaptive Positioning and Localisation in GNSS-Denied Scenarios—A Systematic Review"

_sensors, 2021, doi:10.3390/s22010110_

Round 1

Reviewer 1 Report

LBS service is a hot topic thesedays, a review work is necessary.

In this work the authors have made lots of work, but the manuscript is not well organized, it is a little hard to follow. Besides, for the title is "Cloud Platforms for Context-Adaptive Positioning", the manuscript seems divergent and not focused enough  on this topic. 

Author Response

Dear Reviewer,

Please find attached the response to the comments provided to our systematic review.

Thank you for your help.

Best regards,

Authors 

Reviewer 2 Report

Dear authors,

Please find the attached file for my comments. Please revise the paper based on the comments and resubmit it.

Best Regards

Author Response

(The authors gave the same response as above.)

Round 2

Reviewer 1 Report

I have no other comments.